# Characterization of Femtosecond Laser and Porcine Crystalline Lens Interactions by Optical Microscopy

**DOI:** 10.3390/mi13122128

**Published:** 2022-12-01

**Authors:** Olfa Ben Moussa, Abderazek Talbi, Sylvain Poinard, Thibaud Garcin, Anne-Sophie Gauthier, Gilles Thuret, Philippe Gain, Aurélien Maurer, Xxx Sedao, Cyril Mauclair

**Affiliations:** 1Laboratory Biology, Engineering and Imaging for Ophthalmology, BiiO, Faculty of Medicine, Health Innovation Campus, Jean Monnet University, 42000 Saint-Étienne, France; 2Laboratoire Hubert Curien, UMR 5516 CNRS, Université Jean Monnet, 42000 Saint-Étienne, France; 3KEJAKO, 42000 Saint-Etienne, France

**Keywords:** ultrafast laser, crystalline lens, histology, time-resolved imagining, cavitation

## Abstract

The use of ultrafast laser pulses for eye anterior segment surgery has seen a tremendous growth of interest as the technique has revolutionized the field, from the treatment of myopia, hyperopia, and presbyopia in the cornea to laser-assisted cataract surgery of the crystalline lens. For the latter, a comprehensive understanding of the laser–tissue interaction has yet to be achieved, mainly because of the challenge of observing the interaction zone in situ with sufficient spatial and temporal resolution in the complex and multi-layered tissue of the crystalline lens. We report here on the dedicated characterization results of the laser–tissue interaction zone in the ex vivo porcine lens using three different methods: in situ and real-time microscopy, wide-field optical imaging, and phase-contrast microscopy of the histological cross sections. These complementary approaches together revealed new physical and biological consequences of laser irradiation: a low-energy interaction regime (pulse energy below ~1 µJ) with very limited cavitation effects and a stronger photo-disruption regime (pulse energy above 1 µJ) with a long cavitation duration from seconds to minutes, resulting in elongated spots. These advances in the understanding of the ultrafast laser’s interactions with the lens are of the utmost importance for the preparation of the next-generation treatments that will be applied to the lens.

## 1. Introduction

Surgery of the anterior eye segment has seen a strong evolution since the first use of ultrafast laser pulses for highly localized photodisruption, which enabled precise and well-controlled cuts of the cornea [1] and, more recently, of the crystalline lens [2]. Ophthalmic applications of ultra-short lasers have been considered since 1989 [3], and the first ophthalmic femtosecond laser (fsL) was commercialized in 2004. Since then, ultrafast lasers have been used to first cut the cornea and then, shortly after, the lens: Kymionis et al. and Shah et al. performed a laser incision in the cornea for refractive correction [4,5]. Palanker et al. cut planes inside the lens to facilitate the ultrasonic phacoemulsification and aspiration steps of the fragmented lens [2]. More recently, Bernard et al. showed the application of ultrafast laser pulses for full-lens photo-emulsification [6].

Nowadays, there are several surgical techniques involving the use of laser pulses along with the fast displacement of the laser spot using scanning mirrors to generate cuts with user-defined geometry. For example, the well-known laser-assisted in situ keratomileusis surgery (LASIK) consists of achieving a large planar cut within the cornea to generate a flap that is manually opened before UV laser tissue ablation for refractive correction [4]. Other cutting geometries are used for small-incision lenticule extraction (SMILE) in the cornea and for refractive correction purposes as well [5]. The same holds true for femtosecond laser-assisted cataract surgery (FLACS), where large cutting planes within the lens are performed to ease the ultrasonic phacoemulsification and aspiration steps of the fragmented lens [2]. More recently, full-lens photo-emulsification by ultrafast laser pulses followed by aspiration (without an ultrasound) has been demonstrated, using arrays of ultrafast laser spots [6]. Studies have been conducted to better understand and control the interaction between the fsL pulse and the transparent biological tissue [6,7]. Most of these studies relied on in situ imaging pump–probe setups to characterize the cavitation and bubble generation, usually in water [8] or in gelatins mimicking some of the physical properties of the biological tissue [9,10].

While these studies have enabled remarkable advances in the understanding of this interaction, even leading to analytical models for cutting optimization [11] and precise evaluation of the generated shock wave in an artificial model eye [12], finer investigations on in vivo or ex vivo animal lenses and/or ex vivo human lenses are still required to fully understand and control this interaction down to the micrometric scale, especially to develop new therapeutic applications, in particular non-destructive ones, and to fully understand the laser–biological-tissue interaction. Indeed, from the biological point of view, the lens is a very complex, multi-layered tissue with elongated cells and various protein aggregates implying various local biological and physical properties [13]. Given the interest in the lens’s aging processes and in strategies to counteract the associated loss of accommodation, eventually leading to presbyopia, it is obvious that investigations involving fine characterizations of the interaction zone in an actual lens are required not only to assess the effects of a single pulse but also, at a larger scale, to evaluate the consequences of several irradiation sites. As part of this trend, the visualization of fsL-generated microincisions in the lens with confocal microscopy has been reported [14], showing a dependence by the local cutting effect on the laser energy. The same group also investigated the possibility to alter ex vivo the lens’s biomechanics by generating several fsL cutting plans inside the lens cortex (i.e., outside the nucleus) in order to soften the lens and treat presbyopia [15].

If these first investigations are clearly on the right path to better discern and master the laser–lens-tissue interaction, more efforts are necessary to reach the full potential of ultrafast laser pulses in treating the lens and, especially, presbyopia, which involves various complex biological mechanisms [16,17]. In addition, real-time in situ observation of the laser-induced cavitation in the lens with a micrometric resolution is still lacking, even though the expansion and cavitation dynamics are presumably different from the commonly used model media such as water or porcine gelatin [8,9,10].

In the present paper, we report on the precise characterization of the laser–tissue interaction zone in the nucleus of ex vivo porcine lenses (embryonic–fetal–infantile and adult nucleus), the biological structure responsible for presbyopia because it hardens progressively with aging [18]. We use three methods at different observation scales, namely in situ real-time microscopy, wide-field optical imaging, and microscopy of the histological cross sections.

These complementary approaches reveal new laser-induced lens modifications. In particular, we identify a new laser-induced lens-modification regime, with low cavitation effects. Finally, we also observe a long laser-induced cavitation duration of 5–10 min.

## 2. Materials and Methods

A group of 20 pig lenses (10 animals aged: 2–6 years) were treated by fsL pulses (Satsuma, Amplitude, Bordeaux, France) at a wavelength of 1030 nm, with a pulse duration of 300 fs at full width half maximum (FWHM) and a Gaussian-shaped spot with a diameter of 5 μm at 1/e^2^. Laser settings were similar to those of commercial lasers used in refractive surgery (for instance, Abbott Intralase 150, Alcon Wavelight FS200, and Zeiss VisuMax).

FsL pulses were focused at specific X and Y coordinates using a scanner and a lens system with submicrometric positioning precision. Moreover, a linear translation stage supporting the porcine lens was used to control the Z position. The laser irradiation pattern consisted of a two-dimensional array in the X–Y plane of single-pulse laser-induced photodisruption (one shot per location) with an inter-site distance of 15 μm, as illustrated in Figure 1. This irradiation site separation of 15 µm is larger than the maximum individual bubble radius observed in water in similar irradiation conditions [19]. It was chosen to limit the possible light deflection caused by transient cavitation bubble on subsequent pulses.

The laser was set at 50 kHz repetition rate, laser power was measured, and then the laser-pulse energy was deduced from power measurements and repetition rate. The pattern was repeated for 7 different pulse energy levels, as shown in Table 1.

The whole procedure was applied at 12 different Z positions with an inter-plane spacing of 500 μm in order to target different areas from the adult to the embryonic nucleus, i.e., layers corresponding to the hardest zones of a presbyopic lens in humans (see Figure 1a). The fsL optical axis was directed along the Z axis, from the posterior face towards the anterior face. As a prompt inspection of laser-treated lens, a binocular microscope (SZ61, Olympus, Tokyo, Japan) was used to image the irradiated zones with a large field of view (magnifications: ×25 and ×45) 5–10 min after laser treatment.

A histological analysis of the irradiated volume was also performed. Briefly, the entire laser-treated lenses were fixed for 24 h in 0.75% paraformaldehyde (PFA) in Dulbecco’s phosphate-buffered saline (DPBS). A thick slice was then performed perpendicularly to the optical *Z*-axis in the posterior face with a sharp blade (described in [20]), in order to reveal the cross section of the laser-impacted zones. The slice was fixed again for 24 h in 0.75% PFA and then incubated in 10% and 20% sucrose–DPBS for 4 h and 30% of sucrose–DPBS for 24 h at +4 °C. Finally, the samples were incubated in the optimal cutting temperature compound (OCT, Thermo Scientific, Runcorn, UK), frozen in liquid nitrogen, and stored at −20 °C. Lenses were cryosectioned into 60 μm thick sections (Leica CM1520 cryostat, Leica Biosystems, Nussloch GmbH, Nußloch, Germany) and transferred onto 20% OCT–DPBS-coated microscope slides (series 2 adhesive microscope slides, Trajan Scientific and medical, Milton Keynes, UK). Laser-disruptions sites were observed using bright-field microscopy (IX81, Olympus, Tokyo, Japan).

As a third characterization means, a real-time microscopy setup based on phase-contrast microscopy with white-light illumination (described in [21]) was used to perform side-view imaging of single-laser pulses’ irradiation (i.e., images in the Z–X plane) inside the ex vivo pig lens. This allowed for recording of an in situ video of the laser interaction at 1.2 μJ in real-time with video frame rate. In this case, the fsL pulses were at a central wavelength of 800 nm, with an FWHM pulse duration of 100 fs and a beam profile of similar size to the aforementioned Gaussian case.

## 3. Results

### 3.1. Wide-Field Transmitted-Light Microscopy of the Whole Lens

Figure 2 presented a typical image of the irradiated zones, as observed by the anterior face of the lens within 5–10 min after laser irradiation.

The large field of view allowed for visualizing of the modification caused by the laser treatment at all energy levels. Three types of interactions were identified, namely (1) marked bubbles with detectable coalescence, (2) small and sparsely located bubbles, and (3) no presence of bubbles but macroscopically opaque. Firstly, numerous gas bubbles were systemically observed in 19 out of 20 lenses in the zones having received a laser energy equal to or above 1 μJ (E4), which corresponded to a mean fluence of about 5 J/cm^2^. The bubbles’ size and density increased with the laser energy. Relatively large bubbles (up to 200 μm in diameter) were observed at 2.3 μJ (E1), as a result of a coalescence phenomenon of the small bubbles due to the small inter-spots space (15 μm). These bubbles remained until the histology sample preparation, which took place 24 h after fsL irradiation (details are presented in Figure 3 and Figure 4). Secondly, for the zones that received laser-pulse energy equal to or below 0.82 μJ (E5), visible/hardly visible or no bubbles were detected in 9 out of 20 lenses. Thirdly, a macroscopic and quite uniform opacification without bubbles was detected at 0.54 µJ (E7) in 4 out of 20 lenses. We also noted visible or hardly visible bubbles for all laser-pulse energy levels in 10 out of 20 lenses. The details of the interactions for each lens are shown in Figure 3.

### 3.2. Phase-Contrast Imaging on Cross Sections

Histology revealed the cross-section pattern of the laser-induced alterations inside the lens nucleus. Well-marked laser impacts were present in the center of the lens (just below the embryonic nucleus) at 2.31 μJ (E1). The gas bubbles seen soon after laser impact (Figure 2) were not visible anymore (resorbed with time and/or because of the sample-cutting procedure). Seventeen elongated and well-separated slight spots were observed for each laser irradiation site, especially those that received a high laser-energy dose from 2.31 µJ (E1) to 1 µJ (E4). Interestingly, the tissue located between the photodisrupted sites seemed intact, as compared to the surrounding unexposed areas, with an apparently preserved lens fibers’ network. Moreover, we observed morphological and optical differences between the laser spots at 2.31 µJ of two adjacent layers in Z (E1), as shown in A and B in Figure 4b. Considering that both areas (A and B in Figure 4b) received the same pulse energy and that these zones were separated by only 500 μm, the observed differences could not have been caused by an increase in spherical aberrations but rather by tissue heterogeneity. Finally, for energy below ~1 µJ, no alteration was visible on the cross sections with phase contrast.

### 3.3. Time-Resolved Imaging

We further characterized the fsL–lens interaction using real-time in situ microscopy to better understand the lens response over time and at the micrometric scale. Figure 5a depicted the experimental imaging setup offering the possibility to visualize the laser–lens interaction in real time and in phase contrast, by recording a video of the laser impact inside the lens using a phase-contrast microscope (the bubbles appeared dark in this microscopy technique). A single laser pulse was applied, and the pulse energy was 1.2 μJ, which was above the bubble’s generation threshold. Time-lapse images over several seconds are presented in Figure 5b. At t = 0 s, a micro-bubble was generated, as observed previously (Figure 2). Under this side imaging, the bubble appeared elongated following the extended laser-intensity distribution along the optical axis. The diameter of the bubble steadily decreased over time and completely disappeared after 130 s. A permanent modification persisted in the last picture (t = ∞), corresponding to the micro-cavitation shown previously with the histological characterization.

## 4. Discussion

Here, we describe the new physical and biological consequences of laser irradiation inside the porcine lens. Three different interaction regimes with permanent lens tissue modifications are identified, namely bubbles formation with consequent coalescence, bubbles formation with negligible coalescence, and no bubbles formation but with opacification. These regimes are clearly correlated to the laser-pulse energy applied. To the best of our knowledge, the latter is a newly identified regime. The opacification is possibly linked to a localized refractive index change related to the local modification of the lens-tissue structure and/or proteins.

For a precise characterization of the laser interaction, we used three different complementary methods: wide-field optical imaging, microscopy on the fine histological cross sections, and in situ real-time microscopy.

Firstly, we used optical imaging in order to image the irradiated lenses with a large field of view and directly compare the effect of the laser energies. This allows for identifying the low-energy interaction regime (0.68–0.54 µJ) that was not observable with the high-resolution histological imaging. In this case, the high density of the laser impacts—that are presumably quasi-undetectable alone—has collectively changed the overall contrast of the laser-irradiated area, making it visible with a low-magnification transmitted-light microscope, despite its limited resolution. This method allows for detecting local modification in the lens, which is a heterogeneous sample composed of four nuclear layers [18]. However, this method of observation permits analysis of only the most superficial areas, since the deepest areas are masked by the upper layers (see Figure 1).

Moreover, we point out here the inter-lenses variability (old breeders, aged 2–6 years), which has a significant effect on the laser–lens interaction, especially at low energy, from 0.82 µJ to 0.54 µJ. Differences in the laser–tissue interactions are also noticeable within the same lens’s nucleus, depending on the area being treated (A and B in Figure 4b). This could be explained by the unique organization of the fiber cells. Morphologic studies conducted with electron microscopy have enabled visualization of the fiber cells’ structure. Indeed, in the cortex, they are arranged in radial cell columns, whereas, in the adult nucleus, the fibers are packed and irregularly shaped. In the infantile nucleus, their shape is similar to the fibers of the adult nucleus, but they are less compacted. In the fetal nucleus, the fibers are rounded and organized in irregular rows. Finally, in the embryonic nucleus, they are irregularly shaped and arranged in a heterogenous pattern [22,23]. Moreover, the lens is a unique tissue because of its water and refractile protein content. An adult human lens capsule is about 80% water; the cortex contains 68.6% water, and the nucleus contains 63.4% water. However, the refractive index of the lens is not uniform: a higher refractive index is present in the nucleus (1.41) than in the cortex (1.38) [23].

At high energy, between 2.31 µJ and 1 µJ, large bubbles (up to 200 μm in diameter) and coalescent ones were initially visualized due to the small inter-spots space (15 μm). This observation corresponded to what was described by Krueger et al. in pig lenses [24], with disrupted membranes and fibers with denser layers at the border of the cavitation bubble. Nevertheless, the closer inspection enabled by the histology cross sections reveals that the tissue surrounding each modification site remains unaffected. Thus, the bubble coalescence phenomenon may occur without a marked and continued tissue disruption throughout the irradiated sites. Further investigations are required to better understand this phenomenon.

Histology revealed laser modifications inside the lens nucleus. At a high laser-energy dose from 2.31 µJ to 1 µJ, elongated and separated slight spots were observed with an apparently preserved lens fibers’ network. Moreover, we observed morphological and optical differences between laser spots at 2.31 µJ. It is still unclear, so far, what the source of the differences is amongst the interaction sites of the same laser condition. We supposed that the tissue inhomogeneities coupled to the complexity of the histological cutting procedure may explain the morphological and optical differences between A and B in Figure 4b. In the future, thinner cross-section and immunohistochemistry techniques will improve laser–lens observation at the cellular scale.

Finally, the real-time imaging setup for the visualization of laser-irradiated lens tissue at a micrometric-scale resolution is also manifested to be a potent tool for a comprehensive understanding of laser–lens interaction being achieved in the near future. Although already being frequently used in laser irradiation studies for material transfers and hydrodynamics [25,26], this technique has rarely been applied to biological materials [27] or, even more rarely, to biological tissue or organs in an ex vivo fashion. Similar to histology, this tool allows for visualizing the volume and geometry of the interaction event.

In addition, the images acquired from this setup yield readily valuable information about the formation and lifetime of the bubbles and the dimension of the permanent trace. The resorption phenomenon seems to occur quickly compared to our previous results, where we could observe the bubbles until 10 min after the laser treatment on the large zones with a strong density of laser impacts (Figure 1). The resorption phenomenon might occur more quickly in the case of a single irradiation site, compared to the cases of high irradiation-density areas. This highlights the interest of employing various characterization means to observe the individual and collective phenomena. In any case, we notice a cavitation lifetime that is significantly longer than what is commonly observed in water or in porcine gelatin [10]. This result is in favor of further studies in actual crystalline lens media with time-resolved in situ characterization at ns–μs scales.

With further improvement of the setup, notably quantitative phase imaging, time-gated pump–probe imaging for plasma, and consequent shock propagation, further understanding is expected to be achieved.

## 5. Conclusions

By combining three microscopic techniques having complementary spatial and temporal resolutions, we report on a low-energy interaction regime, highly dependent on the irradiation lens area, with low cavitation effects. We also observe permanent elongated traces associated with a long laser-induced cavitation duration up to 10 min. Further studies involving in situ observation with a time resolution on the order of ns and analysis of the histological and biochemical changes are needed to understand this new laser–crystal interaction regime.

## Figures and Tables

**Figure 1 micromachines-13-02128-f001:**
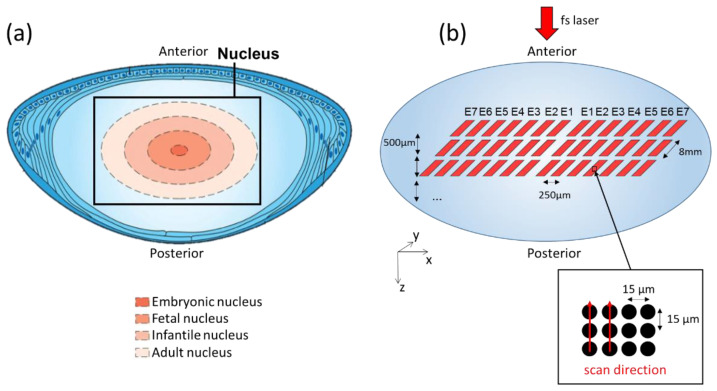
Femtosecond laser pattern on the crystalline lens nucleus. (**a**) Diagram of porcine crystalline lens anatomy (adapted from Ento Key, basic science of lens, https://entokey.com/basic-science-of-the-lens/, accessed on 3 October 2019) showing, on a cross section, the 4 layers from the adult to the embryonic nucleus; (**b**) 3D schematic of the single-pulse laser patterns with various energies and depths for the investigation of laser–lens interaction within the nucleus volume.

**Figure 2 micromachines-13-02128-f002:**
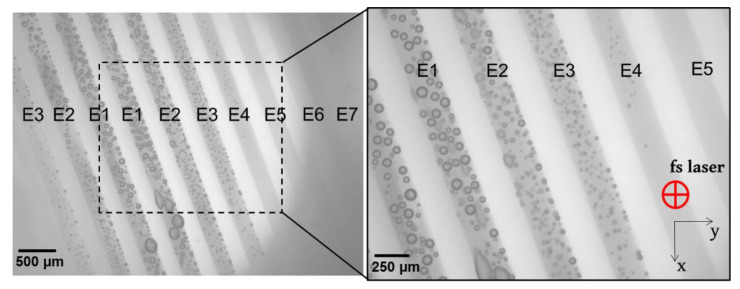
Top-view of one laser-treated ex vivo pig lens after exposition to different femtosecond laser energies. Images were acquired 5 min after irradiation, using an optical microscope at ×25 and ×45 magnifications. The laser settings are explained in Figure 1. Microscopic top-view provides information about the upper-most layer of laser-treated matrices. Cavitation and coalescence of bubbles are perfectly visible from high laser-pulse energy settings.

**Figure 3 micromachines-13-02128-f003:**
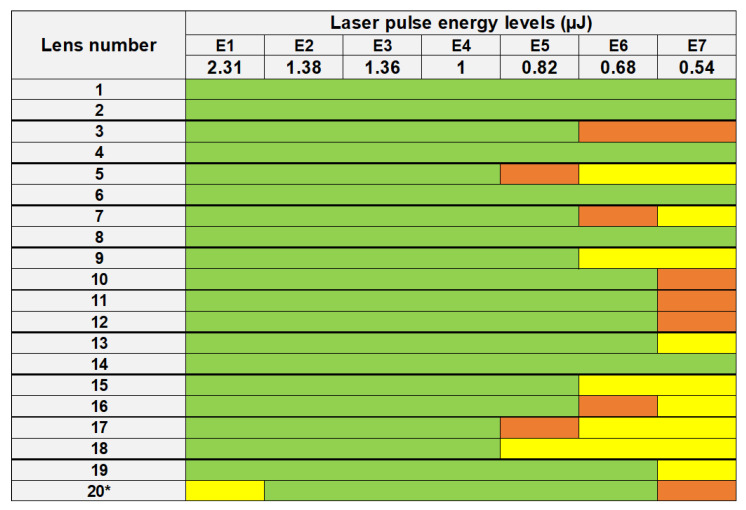
List of lenses treated by laser-pulse energy levels from 2.31 µJ (E1) to 0.54 µJ (E7). Marked bubbles with detectable coalescence (green), small and sparsely located bubbles (orange), and no bubbles but macroscopically opaque (yellow) are shown for each lens. A quite uniform opacification without bubbles was observed at 2.31 µJ (E1) for lens no. 20*. Pairs of lenses are separated by bold lines.

**Figure 4 micromachines-13-02128-f004:**
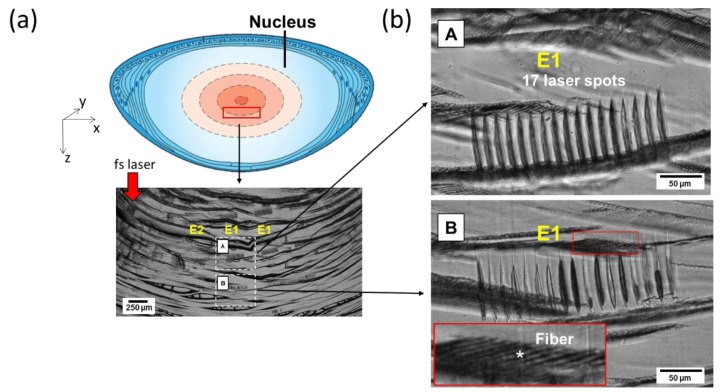
Microscopic images of 60 μm thick histological slices of typical laser-irradiated zones. (**a**) Scheme of the observed area within the lens with an overview of the various laser-impacted zones’ cross sections. (**b**) Magnified view of two zones of interest (**A** and **B**) exposed to the same laser-pulse energy and at a similar X–Y position within the lens but at different Zs. Wherever the irradiation sites are clearly observable and well contrasted, the fiber network (*) appears relatively unaffected.

**Figure 5 micromachines-13-02128-f005:**
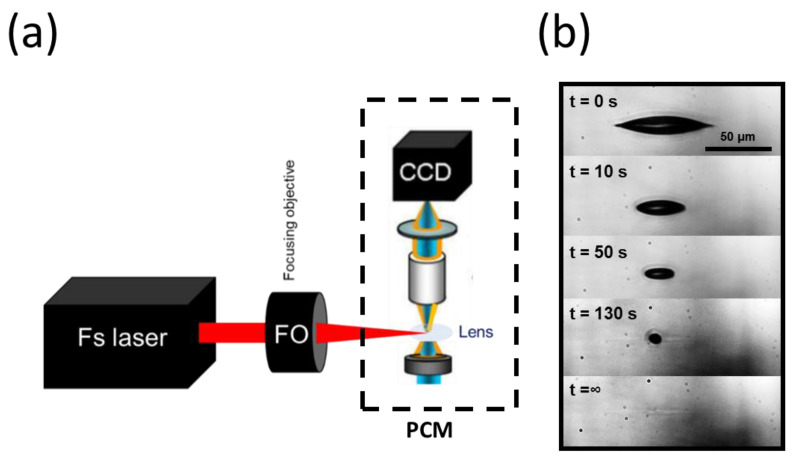
Time-resolved imaging device for real-time visualization of the cavitation. (**a**) Diagram showing the experimental setup for real-time visualization of the cavitation. The system was based on a conventional phase-contrast microscope (PCM), on which the femtosecond laser beam was focused onto the ex vivo pig lens. With fine adjustments, the irradiation zone was brought under the microscope’s field of view. A real-time video microscopy of the laser–tissue interaction was captured by a CCD camera with high spatial resolution and phase sensitivity (IXon, Andor Technology, Abingdon, UK). (**b**) A selection of real-time images acquired during the cavitation process in the lens. The time stamps are marked on the top-left of each image (with s for seconds). The bubble lifetime was about 130 s, after which a slight elongated trace persisted (enhanced contrast).

**Table 1 micromachines-13-02128-t001:** Laser-pulse energy levels used to treat ex vivo pig lenses.

Pulse Energy	E1	E2	E3	E4	E5	E6	E7
µJ	2.31	1.38	1.36	1	0.82	0.68	0.54

## Data Availability

The data that support the findings of this study are available from the corresponding author upon reasonable request.

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
