# Peer review of "Characterization of Femtosecond Laser and Porcine Crystalline Lens Interactions by Optical Microscopy"

_micromachines, 2022, doi:10.3390/mi13122128_

Round 1

Reviewer 1 Report

The authors reported to irradiate crystalline lens of pig eye by femtosecond laser, time-resolved microscopy, wide field optical imaging, and phase contrast microscopy have used to investigated the affected aero. Laser energy depended changes have been classified into three types. It has potential use on laser-assisted cataract surgery and eye anterior segment surgery by ultrafast laser. Some suggestions and questions:   

1, the authors claim opacification without bubbles formation to be a newly identified regime. It is like the haze after Laser-Assisted in situ Keratomileusis Surgery (LASIK) by UV laser. It is better to compare the similar effects of fs laser and UV ns laser.

2, what’s different between two kinds of bubble in Figure2 and in Figure5? One is transparent, the latter is black.

3, what is look like Figure2 under phase contrast microscopy?

Author Response

Response 1: we thank the authors for this very interesting question. If much more investigations are required to clearly address this point, it is the authors opinion that the interaction is presumably quite different between the cornea and the crystalline lens. First, not only the chemical components involved in these two tissues strongly differ but also the cell types, size, organization and function (cell layers, hydration, transparency, ageing mechanisms and so on). If the observed opacification regime in the lens shows similarities in terms of local opacity with the LASIK haze, a first obvious difference is the delay of appearance. The haze is observed days/weeks after the LASIK surgery [1] whereas we have observed the opacification right after the laser treatment in the lens. Lastly, the interaction in the case of UV ns irradiation relies on linear absorption tissue is absorptive whereas the femtosecond pulses deliver photons on a much shorter timescale with absorption mechanisms relying on nonlinear absorption, see the reference chapter from Vogel et al. [2].

Response 2: we used two different imaging techniques : wide field transmitted light microscopy for Figure 2, which gives transparent bubbles, and phase contrast microscpy for Figure 5, giving dark bubbles. 

Response 3: in the future, it will be interest to observe Figure 2 under phase contrast microscopy. Phase contrast can very well complete the characterization when associated with a normal transmission microscopy yielding the cartography of both absorption and refractive index variations.

References

  1. Aleksandar Stojanovic, Tore A Nitter. Correlation between ultraviolet radiation level and the incidence of late-onset corneal haze after photorefractive keratectomy. Journal of Cataract & Refractive Surgery,Volume 27, Issue 3, 2001, Pages 404-410, ISSN 0886-3350, doi: 10.1016/S0886-3350(00)00742-2.
  2. Vogel, A.; Venugopalan, V. Mechanisms of Pulsed Laser Ablation of Biological Tissues. Chemical Reviews 2003, 103, 577-644. Publisher: American Chemical Society, doi:10.1021/cr010379n.

Reviewer 2 Report

In this paper, three microscopic techniques are applied to study the interactions between femtosecond laser and the pig crystalline lens, which is of great interest to the readers. I think it could be accepted after a minor revision.

Here are some comments:

1. Could you give some comments on the bubbles generated in the irradiation experiment? Is it a detrimental phenomenon? Does it affect the laser surgery on the crystalline?

2. The first paragraph of the discussion session can be moved to the introduction session since it is more about research progress rather than a summary.

3. Be careful of some units, there should be some spaces between the numbers and them.

4. What is the repetition rate of the femtosecond laser? As one of the important parameters of the femtosecond laser, it also should have some impact on the interactions between the laser and the crystalline lens. And as you know, the higher the repetition rate, the higher the average power, so totally you cannot ignore this parameter.

Author Response

Response 1: knowing the bubble dynamics in terms of time and space is of outmost importance in order to avoid any shadowing of light deflection effect in the case of multi-pulse irradiation inside transparent tissues. It is also needed to evaluate the resorption phenomena of the irradiated site as this resorption is ultimately needed to preserve the lens transparency. The bubble generated can have several detrimental effects, two of them must draw our attention:

  1. During the surgery of the transparent tissue, the focused laser pulse must interact with a fresh area exempt of transient bubbles from the previous pulses. These could shadow and/or deflect the optical rays and thus render the surgery unstable and not controlled. Thus, several investigations have been conducted to evaluate the bubble lifetime in water, cornea [3] or liquid with porcine gelatine [4]. A bubble lifetime up to 10 µs with a maximum diameter of 25µm is observed in water for irradiation conditions similar to the one employed in our study [5].This means that a repetition rate lower than 100 kHz and a spatial separation of 12.5 µm permits to limit detrimental optical effects on the incoming laser pulse due to the vanishing bubble. Our experimental conditions with a repetition rate of 50 kHz and a spot separation of 15 µm follow these requirements. Now, as we observed much longer bubble life time than in water so 15 µm is the spatial separation of the irradiation site which was necessary to limit this effect;
  2. Another potential detrimental effect is the opacification of the irradiated area that, if permanent, would render the surgery too risky. These areas must also recover transparency. Resorption of femtosecond laser photodisruption zones has been observed on rabbit lenses in a similar experiment performed in vivo where gliding planes were achieved with a large onset of irradiation sites forming radial planes (Lentotomy) [6]. Naturally a dedicated thorough study of the resorption phenomena is out of the scope of this report and will have to be conducted, but it is clearly a resorption phenomenon for surgery: further investigation nedded is underway.

Response 2: we thank the reviewer for this comment. The first paragraph of the discussion session was moved to the introduction session (see lines 35-42).

Response 3: we placed spaces between the numbers and their units.

Response 4: the laser was set at 50 kHz repetition rate, laser power was measured, then the laser pulse energy was deduced from power measurements and repetition rate. This information was placed in the new version of our article (see lines 108-109).

References

  1. Juhasz T, Kastis GA, Suárez C, Bor Z, Bron WE. Time-resolved observations of shock waves and cavitation bubbles generated by femtosecond laser pulses in corneal tissue and water. Lasers Surg Med. 1996;19(1):23-31. doi: 10.1002/(SICI)1096-9101(1996)19:1<23::AID-LSM4>3.0.CO;2-S.
  2. Tinne, N.; Kaune, B.; Krüger, A.; Ripken, T. Interaction Mechanisms of Cavitation Bubbles Induced by Spatially and Temporally Separated fs-Laser Pulses. PLoS ONE 2014, 9. doi: 10.1371/journal.pone.0114437.
  3. Alberto Aguilar, Aurélien Bernard, Amélie De Saint-Jean, Emmanuel Baubeau, Aurélien Bertail, and Cyril Mauclair, "Astigmatism and spherical aberrations as main causes for degradation of ultrafast laser-induced cavitation in water," OSA Continuum 4, 2905-2917 (2021)
  4. Holger Lubatschowski, Silvia Schumacher , Michael Fromm et al. Femtosecond lentotomy: generating gliding planes inside the crystalline lens to regain accommodation ability. J. Biophoton. 3, No. 5–6, 265–268 (2010) / doi: 10.1002/jbio.201000013

Reviewer 3 Report

The topic of this article is very interesting and relevant. The study of the effect of femtosecond laser radiation on the lens of the eye is promising for the further development of medicine. Determining the effect of pulse energy on the lens tissue is very interesting. But there are some notes:

1. In Section 3.3, the description does not clearly state that bubble relaxation occurs after exposure to femtosecond pulses or during prolonged exposure to laser radiation. It can be understood from the text that the laser exposure takes 130 seconds.

2. Do the bubbles that form at 2.3 microjoules (the experiment described in section 3.1) disappear over time, as they do in section 3.3?

3. In line 242, the unit of measure (microjoule) has gone to another line, and there are no problems anywhere between the unit of measure and the value.

Author Response

Response 1: we thank the reviewer for this valuable comment. It is indeed a single pulse irradiation, we have modified the text as follows: “A single laser pulse was applied and the pulse energy was   1.2 μJ which was above the bubble’s generation threshold” (see lines 206-207).

Response 2: no, they remain and do not resorb as demonstrated in section 3.2. The same lens described in 3.1 (figure 2) is used in 3.2 (figure 4)

Response 3: we corrected this mistake in the new version of the article.
